# Antioxidant Enzyme System Modulation by Dietary Palm Oils, Palm Kernel Oil and Soybean Oil in Laying Hens

**DOI:** 10.3390/ani13142245

**Published:** 2023-07-08

**Authors:** Wan Ibrahim Izuddin, Teck Chwen Loh, Nazri Nayan, Henny Akit, Hooi Ling Foo, Ahmadilfitri Md Noor

**Affiliations:** 1Department of Animal Science, Faculty of Agriculture, Universiti Putra Malaysia, Serdang 43400, Malaysia; wanahmadizuddin@gmail.com (W.I.I.); nazri.nayan@upm.edu.my (N.N.); henny@upm.edu.my (H.A.); 2Institute of Tropical Agriculture and Food Security (ITAFoS), Universiti Putra Malaysia, Serdang 43400, Malaysia; 3Institute of Bioscience, Universiti Putra Malaysia, Serdang 43400, Malaysia; hlfoo@upm.edu.my; 4Department of Bioprocess Technology, Faculty of Biotechnology and Biomolecular Science, Universiti Putra Malaysia, Serdang 43400, Malaysia; 5Sime Darby Plantation Research Sdn Bhd, R&D Centre—Carey Island, Lot 2664, Jalan Pulau Carey, Carey Island 42960, Malaysia; ahmadilfitri.md.noor@simedarbyplantation.com

**Keywords:** crude palm oil, red palm oil, refined palm oil, palm kernel oil, soybean oil, antioxidant enzyme activity, superoxide dismutase, glutathione peroxidase, catalase

## Abstract

**Simple Summary:**

Different sources of oils have unique characteristics, such as fatty acids and phenolic compounds. Different compounds in oils have different antioxidant capacities, and their inclusion in the diet would influence the antioxidant system in poultry. In comparing dietary supplementation of palm oils (crude palm oil, red palm oil, refined palm oil), palm kernel oil and soybean oil in laying hens, we found that palm oils with high antioxidant content had a lower production of antioxidant enzymes in the intestinal mucosa, serum and liver. We suggest that the inclusion of oils with high antioxidant content, such as crude palm oil and red palm oil, provides better protection through reduction in oxidation.

**Abstract:**

Palm-based oils (palm oil and kernel oil) and soybean oil have unique fatty acid and antioxidant profiles based on the compounds present in them. Hence, this study elucidated the antioxidant properties of crude palm oil (CPO), red palm oil (RPO), refined palm oil (RBD), palm kernel oil (PKO) and soybean oil (SBO) and the influence of dietary oils on blood lipid profiles, tissue fatty acid deposition and the expression of hepatic lipid and lipoprotein metabolism genes in laying hens. The oils were analyzed for color, beta-carotene, free fatty acid and acid value, phenolic content and lipid peroxidation. In an in vivo trial, 150 laying hens were allotted into five groups and supplemented with either CPO, RPO, RBD, PKO or SBO for 16 weeks. High antioxidant compounds present in palm oils help reduce the oxidation of oils. Dietary supplementation with palm oils, particularly CPO and RPO, contributed to the lower liver, serum and jejunal mucosal antioxidant enzyme activities. The antioxidant enzyme genes in the jejunal mucosa were downregulated in palm oils and PKO, but there was no difference between oils in antioxidant enzyme genes in the liver. In conclusion, dietary supplementation with oils with high antioxidant content contributed to protection against oxidation and was associated with a lower requirement for producing antioxidant enzymes.

## 1. Introduction

Soybean, palm, rapeseed (canola) and linseed oils are commonly used in poultry feed [1], and palm oil is the cheapest source of oil in the market. As of the first quarter of 2023, the average price for crude palm oil was MYR 3909/ton, RBD palm olein was MYR 4140/ton, crude palm kernel oil was MYR 3865/ton (Malaysian Palm Oil Board) and soybean oil (SBO) was around MYR 5498/ton (World Bank). Including palm oil and palm kernel oil (PKO) in the poultry diet is more cost-effective as it could deliver similar productivity compared to SBO. Dietary supplementation of palm oil, PKO and SBO in laying hens had similar laying performance in terms of hen-day egg production, egg weight, feed intake and feed conversion ratio [2,3].

Palm oil has a balanced fatty acid composition where the fraction of saturated fatty acids (SFA) is almost proportional to the fraction of unsaturated fatty acids (USFA). Palmitic acid (44–45%) and oleic acid (39–40%) are the dominant SFA and USFA, respectively, together with linoleic acid (10–11%) and a minute amount of linolenic acid [4]. PKO is rich in medium-chain fatty acids (MCFA), mainly lauric (C12:0) and myristic (C14:0) acids and contains up to 80% SFA. SBO is polyunsaturated fatty acid (PUFA)-rich oil extracted from soybean and undergoes refining, bleaching and deodorization to produce refined SBO. Table 1 summarizes the fatty acid and saturation profiles of the oils [3]. 

In addition, palm oil has beneficial components present in small amounts, such as carotenoids responsible for the reddish-orange color and vitamin E in the form of tocopherols and tocotrienols. Tocotrienols are potent antioxidants at higher levels in palm oils than in other vegetable oils [5]. Figure 1 summarizes the fatty acid, carotenoids and vitamin E properties of the oils. The oxidation of lipids during processing and storage is caused by variations in chemical reaction mechanisms, including autoxidation and photosensitized oxidation. The oxidative stability of oils is the ability of oils to resist oxidation at processing and storage, which can be described as the periods to achieve the critical point of oxidation, which can be a sensory change or acceleration of the oxidative process [6,7]. Determining oxidative stability is a crucial indicator of oil quality and shelf life, where oxidized oils lead to less acceptable or unacceptable oil for both consumers in household use and industrial use as food ingredients [6,8]. The oxidation of oils affects the essential fatty acids, particularly polyunsaturated fatty acids (PUFA), and generates toxic compounds and oxidized polymers as oxidation products [6].

The fatty acid composition of the oil is a vital factor in determining oxidative stability [9]. The higher the amount of USFA fraction in oil, the greater and faster the oxidation process occurs compared to an SFA fraction in oil [10]. PUFAs, such as linolenic and linoleic acids, have a higher oxidation rate than monounsaturated fatty acids (MUFA), such as oleic acid. Free fatty acids (FFA) are another important criterion in determining the stability of oils. FFA are susceptible to autoxidation compared to esterified fatty acids in triglyceride form, making them a prooxidant in oils [6,11]. Therefore, the presence of high FFA in oils affects their stability. In preventing FFA formation, sterilizing fresh fruit bunches during milling deactivates the hydrolytic enzymes involved and accelerates the reaction that produces FFA. Crude oils contain some amount of FFA, and the refining process removes it to maintain oil stability.

Diet has an important role in modulating gut health by maintaining microbial balance and preventing oxidative stress in the gut. Oxidation is a normal process in the cells. Still, the production of free radical compounds and imbalance in the anti-oxidation process leads to excess radical compounds, causing oxidative stress, which is detrimental to the cells [12]. Antioxidants are required to prevent oxidation by controlling the free radical’s formation level and safely eliminating it from the cells. Antioxidant compounds may be acquired from the diet or antioxidant enzymes produced by the body’s cells. There are several antioxidant enzymes produced by the body, of which superoxide dismutase (SOD), catalase (CAT) and glutathione peroxidase (GPX) are the main antioxidant enzymes responsible for the detoxification of free radicals [12,13]. The SOD enzyme converts superoxide radicals into hydrogen peroxide and oxygen, whereas CAT and GPX convert hydrogen peroxide into water [14]. The antioxidant enzymes convert superoxide radicals and hydrogen peroxide into water and oxygen as final products and safely eliminates them from the body. Oils containing medium-chain fatty acids (MCFA) had higher antioxidant activity [15]. Higher saturated fatty acids (SFA) and antioxidant compounds, such as carotenes and tocotrienols in palm oils, including crude palm oil (CPO) and red palm oil (RPO), are expected to reduce oxidation in the body. A lower oxidation level would reduce the body’s requirement and production of antioxidant enzymes [13,16].

Palm oils contain natural and potent antioxidants in the form of β-carotene and vitamin E. Dietary palm oil containing carotenoids and vitamin E and dietary palm kernel oil (PKO) having a high fraction of medium-chain fatty acids (MCFA) are expected to modulate the gut lumen environment and affect antioxidant enzyme regulation, mucosal immune response and barrier function in laying hens. Several studies showed that dietary polyunsaturated-rich fatty acids (PUFA) were beneficial to gut health and improved immune status in poultry [17,18,19].

Despite the excellent properties and benefits of naturally occurring antioxidant compounds in palm and kernel oils, no studies have explored their potential, particularly in investigating their dietary influence on antioxidant enzyme systems and lipid peroxidation in poultry. Hence, this study investigated the antioxidant properties of CPO, RPO, refined palm oil (RBD), PKO and SBO and their dietary effects on antioxidant enzyme regulations in the liver, serum and intestinal mucosa. In the current study, SBO containing high USFA was used as a control to palm and kernel oils for the measurement of the relative expression of genes.

## 2. Materials and Methods

### 2.1. Source of Oils

Five oils, namely CPO, RPO, RBD, PKO and SBO, were obtained from Palma Tech Product Sdn. Bhd, Shah Alam, Malaysia. The CPO, RPO and RBD are oils extracted from the palm fruit’s mesocarp and are referred to as palm oils. PKO is the oil extracted from the palm fruit kernel, whereas SBO is the refined oil extracted from soybeans. On receiving day, the oils were mixed well, subsampled into an amber Schott Duran^®^ glass bottle, kept at 4 °C in a chiller and protected from light for subsequent analysis.

### 2.2. Color, β-Carotene, Free Fatty Acid and Acid Value Determination of Oils

The oil color (L, a* and b*) was determined by a ColorFlex EZ spectrophotometer (HunterLab, Reston, VA, USA), as described by Izuddin et al. [2]. The β-carotene was extracted from oil samples using acetone, as described by Biswas et al. [20], and the absorbance of β-carotene was measured at 449 nm using Multiskan™ Go UV/VIS spectrophotometer (Thermo Scientific, Waltham, MA, USA). Free fatty acid (FFA) and acid value (AV) were determined according to Japir et al. [21]. A total of 5 g of oil sample was mixed with 50 mL of pre-neutralized isopropanol (Merck, Darmstadt, Germany) in a 100 mL conical flask. The mixture was added to 500 µL of 1% (*w*/*v*) phenolphthalein indicator and heated to 40 °C. The mixture was titrated with 0.1 N sodium hydroxide (Merck, Darmstadt, Germany) until a pink color formed for at least 30 s. The FFA and AV were calculated according to the following formula:$$\%\;FFA\;as\;palmitic\;acid=\frac{\left(mL\;of\;titrant\times N\;of\;titrant\right)\times 25.6}{weight\;of\;sample\;in\;g}\times \frac{mg\;NaOH}{g\;sample}$$
where 25.6 is the formula for FFA determination and the equivalent factor for palmitic acid is as follows:$$AV=FFA\left(\%\right)\times 2.19$$
where 2.19 is the conversion factor for palmitic acid.

### 2.3. Phenolic, Flavonoid, Tannin and Lipid Peroxidation Determination of Oils

Total phenolic content (TPC) in selected oils was determined by the modified protocol by Severo et al. [22] and Ma et al. [23]. TPC was determined based on the gallic acid equivalent. First, a total of 25 µL of oil or gallic acid standard (0 to 200 μg/mL gallic acid) was mixed with 25 µL of 25% (*v*/*v*) Folin–Ciocalteu’s phenol reagent (Merck, Darmstadt, Germany) and 200 µL of deionized water and incubated for 5 min at room temperature. Next, the mixture was added to 25 µL of 10% (*w*/*w*) sodium carbonate (Merck, Darmstadt, Germany) and incubated in the dark for 60 min at room temperature. The absorbance was measured at 764 nm using a SPECORD^®^ 250 PLUS UV/Vis spectrophotometer (Analytic Jena, Jena, Germany). TPC was calculated based on the gallic acid standard curve (0 to 200 μg/mL gallic acid) and expressed as mg of gallic acid equivalent per gram of oil sample on a fresh basis (mg GAE/g oil).

The total flavonoid compound (TFC) was determined by the aluminum chloride protocol outlined by Gouveia and Castilho [24] and modified by Ma et al. [23]. TFC was determined based on the quercetin equivalent. A total of 80 μL oil sample or quercetin standard (0 to 50 μg/mL quercetin) was mixed with 80 μL of 2% (*w*/*v*) ethanolic aluminum chloride and 120 µL of 50 mg/mL sodium acetate. The mixture was then incubated in the dark for 60 min at room temperature. The absorbance of the oil sample or quercetin standard mixture was measured at 440 nm using a SPECORD^®^ 250 PLUS UV/Vis spectrophotometer (Analytic Jena, Jena, Germany). The TFC was quantified from the constructed standard curve of quercetin (0 to 50 μg/mL quercetin) and expressed as quercetin equivalent per gram of oil sample on a fresh basis (mg QE/g oil).

The total tannin compound (TTC) of oils was determined by the modified protocol of Ma et al. [23], from Zou et al. [25], based on the catechin equivalent. A total of 25 µL sample or standard (0–1000 μg/mL catechin) was mixed with 150 µL of 4% (*w*/*v*) methanolic vanillin solution and 25 µL of 32% (*v*/*v*) methanolic sulfuric acid and incubated at 25 °C for 15 min. The absorbance of the mixture was measured at 500 nm using a SPECORD^®^ 250 PLUS UV/Vis spectrophotometer (Analytic Jena, Jena, Germany). The TTC was calculated based on a standard curve of catechin (0–1000 μg/mL catechin) and expressed as catechin equivalent per gram of oil sample on a fresh basis (mg CE/g oil).

Lipid peroxidation was determined using a thiobarbituric acid reactive substance (TBARS) assay by measuring malondialdehyde (MDA) as the product of lipid peroxidation in oil samples, according to Cao et al. [26]. A total of 1 g of oil sample was mixed with 5 mL of 10% (*w*/*v*) trichloroacetic acid (Merck, Darmstadt, Germany) and 80 µL butylated hydroxytoluene (Sigma-Aldrich, MI, USA) and shaken in a water bath for 30 min. The solution was filtered with double filter paper to eliminate the remaining oil. A total of 1 mL of the solution was mixed with 1 mL of freshly prepared 0.02 M thiobarbituric acid (Sigma-Aldrich, MI, USA) (in 20% (*v*/*v*) acetic acid) and placed in a water bath at 90 °C for 40 min. Chloroform was added and mixed before centrifuging at 3000× *g*. The chloroform layer was collected into a cuvette, and absorbance was read at 532 using a SPECORD^®^ 250 PLUS UV/Vis spectrophotometer (Analytic Jena, Jena, Germany). The lipid peroxidation was calculated from the linear regression derived from a standard curve of MDA and expressed as the concentration of MDA per gram of oil sample.

### 2.4. Animal, Management and Dietary Treatments of Feeding Trial

The trial in laying hens was performed at the Department of Animal Science’s Poultry Unit, Faculty of Agriculture, Universiti Putra Malaysia. Two tiers of A-type battery cages with individual cages (50 cm depth, 40 cm height and 30 cm width) were used in an open-sided open-house system. The hens received 16 h of light (12 h of daylight and 4 h of additional LED light) and 8 h of darkness. A total of 150 Hisex Brown laying hens were randomly allotted into five groups. Each group had six biological replicates and five hens per biological replicate (30 hens in total per group). The diets were formulated to be isocaloric and isonitrogenous and meet the nutrient requirements of Hisex Brown laying hens. The dietary treatments were a mash diet with 3% oil, either CPO, RPO, RBD, PKO or SBO (Table 2), and fed at 120 g feed/hen daily, as recommended by Hisex Brown nutritional guidelines. Feed was offered daily at 0700, and ad libitum water was supplied through nipple drinkers. The trial was conducted for 16 weeks, from the 22nd to the 38th week of age.

### 2.5. Sample Collection and Analysis

Six hens per treatment group were randomly selected for sacrifice and sample collection. At bleeding point, blood was collected into the tube (BD Vacutainer^®^ Serum Tubes). Blood was allowed to clot on ice, centrifuged (3000× *g* for 15 min at 4 °C) to collect serum and kept at −80 °C. At evisceration, a lower right lobe portion of the liver and the jejunal tract were collected. Three centimeters of the middle part of jejunal tissues were collected, flushed with PBS buffer solution (pH 7.4), cut to reveal the lumen and scrapped with a glass slide to collect the mucosal layer. The liver tissues and jejunal mucosa were snap frozen and kept in a −80 °C freezer until further analysis. A summary of sample collection and analysis is shown in Figure 2.

### 2.6. Serum, Liver and Intestinal Mucosa Antioxidant Enzymes Activity

Liver, serum and jejunal mucosa content of CAT, GPX and SOD were determined using EnzyChrom™ Catalase, EnzyChrom™ Glutathione Peroxidase and EnzyChrom™ Superoxide Dismutase assay kits (BioAssay Systems, Hayward, CA, USA), respectively, according to the protocol supplied. The detailed protocols of the assays were described by Azizi et al. [28]. The CAT assay was based on the CAT degradation of H_2_O_2_ using a redox dye. The difference in the absorbance at 570 nm was directly proportional to the CAT activity in the sample. The GPX assay kit measured the GPX activity based on the NADPH consumption in the enzyme-coupled reactions. The decrease in the absorbance at 340 nm was directly proportional to the GPX enzyme activity in the sample. The SOD assay kit determined the SOD activity by forming a colored product from the reaction of superoxide (O^2−^) (provided by xanthine oxidase) with a WST-1 dye. The SOD enzyme scavenges the O^2−^, reducing the O^2−^ content for the chromogenic reactions, whose color intensity was measured at 440 nm. All the assay absorbance was measured using a Multiskan™ Go spectrophotometer (Thermo Scientific, Massachusetts, USA).

### 2.7. Liver and Intestinal Mucosa Antioxidant Enzymes Gene Expression

Total RNA was extracted using the NucleoSpin^®^ RNA plus kit (Machery Nagel, Dueren, Germany) and transcribed to cDNA using the cDNA Synthesis Kit (Biotechrabbit, Berlin, Germany), outlined by the manufacturer. The target genes were quantified using 4× CAPITAL^TM^ qPCR Green Master Mix (Biotechrabbit, Berlin, Germany) on a LightCycler^®^ 480 instrument (Roche, Basel, Switzerland). The primer information for target and housekeeping genes is described in Table 3. The detailed extraction, reverse transcription and qPCR protocols were described by Izuddin et al. [2]. The target genes are expressed as a fold change of the treated group to the control group (SBO) using the 2^−ΔΔCt^ method [29].

### 2.8. Experimental Design and Statistical Analysis

The feeding trial was subjected to a completely randomized design (CRD). The data analysis was performed on the SAS software package, version 9.4 (SAS Inst. Inc., Cary, NC, USA). The normality of the data was determined using PROC UNIVARIATE based on Shapiro–Wilk. All data obtained were normally distributed. A one-way analysis of variance (ANOVA) through the General Linear Model (GLM) package was used to determine significance. Duncan’s multiple-range test was used for the post-hoc test to compare the treatment means. The difference was deemed significant at *p* < 0.05.

## 3. Results

### 3.1. FFA, AV, β-Carotene and Color of Oils

There were significant differences (*p* < 0.05) between oils in all the parameters (Table 4). The RPO (8.787% FFA, 19.25 mg KOH/g AV) showed the significantly highest (*p* < 0.05) FFA and AV, followed by the CPO (8.107% FFA, 17.75 mg KOH/g AV), PKO (4.610% FFA, 10.09 mg KOH/g AV), RBD (2.050% FFA, 4.490 mg KOH/g AV) and SBO (0.770% FFA, 1.680 mg KOH/g AV). The highest (*p* < 0.05) β-carotene was observed in CPO (4.073 µg/mL) and RPO (3.800 µg/mL) compared to RBD (0.000 µg/mL), PKO (0.0032 µg/mL) and SBO (0.000 µg/mL). 

The lightness of oil was significantly the highest (*p* < 0.05) in PKO (41.28), followed by CPO (31.51), RBD (18.02), SBO (18.02) and lastly, RPO (16.83). The CPO (25.61), RPO (21.10) and PKO (1.584) showed a significantly higher (*p* < 0.05) redness in color, while the SBO (−1.726) and RBD (−2.642) showed greenness. The yellowness was significantly higher (*p* < 0.05) in CPO (49.42), followed by PKO (43.63), RPO (27.02), RBD (10.24) and SBO (4.764).

### 3.2. Total Phenolics, Flavonoids, Tannin and TBARS of Oils

There were significant differences (*p* < 0.05) between different oil types on all the parameters (Table 5). The significantly highest (*p* < 0.05) TPC was in CPO (333.8 µg GA/mL) and RPO (321.1 µg GA/mL), followed by RBD (239.0 µg GA/mL), PKO (122.2 µg GA/mL) and SBO (11.54 µg GA/mL). The significantly highest (*p* < 0.05) TFC was observed in CPO (232.9 µg R/mL), followed by RPO (162.8 µg R/mL), PKO (111.5 µg R/mL), RBD (65.77 µg R/mL) and SBO (36.98 µg R/mL). The CPO (569.7 µg C/mL) and RPO (574.3 µg C/mL) recorded the significantly highest (*p* < 0.05) TTC, followed by PKO (285.6 µg C/mL), RBD (129.7 µg C/mL) and SBO (82.27 µg C/mL). The significantly highest (*p* < 0.05) TBARS value was in SBO (56.56 µg MDA/g), followed by PKO (31.30 µg MDA/g), RPO (14.81 µg MDA/g) and CPO (10.95 µg MDA/g), with the lowest in RBD (6.568 µg MDA/g).

### 3.3. Liver Antioxidant Enzymes

There were significant differences (*p* < 0.05) between oils in SOD, GPX and CAT activity (Figure 3). The PKO and SBO contributed to the significantly highest (*p* < 0.05) SOD; CPO had the lowest (*p* < 0.05) SOD, but the enzyme activity did not significantly differ (*p* > 0.05) between RPO and RBD. The SBO had the significantly highest (*p* < 0.05) GPX in comparison to other sources of oils. The GPX of palm oils (CPO, RPO and RBD) was not different (*p* > 0.05). However, an opposite trend was shown in CAT, where SBO had significantly (*p* < 0.05) lower enzyme activity compared to others. The oil palm sources did not remain significantly different (*p* > 0.05) in the CAT.

### 3.4. Serum Antioxidant Enzymes

There were significant differences (*p* < 0.05) in GPX and CAT but no differences (*p* > 0.05) in SOD between oil types (Figure 4). Serum GPX had significantly higher (*p* < 0.05) activity in SBO, with no difference (*p* > 0.05) from CPO. A significantly lower (*p* < 0.05) GPX was observed in PKO, with no difference (*p* > 0.05) compared to RPO and RBD. There was a significantly higher (*p* < 0.05) CAT serum in RBD, with no difference (*p* > 0.05) to SBO. There was a significantly lower (*p* < 0.05) CAT in CPO and RPO than in RBD and SBO.

### 3.5. Intestinal Mucosa Antioxidant Enzymes

Significant differences (*p* < 0.05) between oils were observed in SOD, but no significant effects (*p* > 0.05) were observed in the GPX and CAT (Figure 5). There was a significantly greater (*p* < 0.05) SOD in RBD and SBO in comparison to RPO, but there was no significant difference (*p* > 0.05) in comparison to CPO and PKO. A significantly lower (*p* < 0.05) SOD was seen in RPO compared to RBD and SBO.

### 3.6. Liver and Mucosa Antioxidant Enzyme Gene Expression

There were no significant effects (*p* > 0.05) of different dietary supplementations of oils on the expression of the liver antioxidant enzyme genes, but significant effects (*p* < 0.05) were pronounced in the jejunal mucosa SOD, GPX and CAT genes (Table 6). The SOD, GPX and CAT enzymes had significantly greater (*p* < 0.05) expression in SBO compared to oil palm groups. The CPO had significantly lower (*p* < 0.05) SOD and GPX than other treatments but was similar (*p* > 0.05) to RPO in SOD and RPO and RBD in GPX. CAT was significantly lower (*p* < 0.05) in CPO and RPO compared to other oils, with no significant difference (*p* > 0.05) in RBD.

## 4. Discussion

### 4.1. FFA, AV and β-Carotene and Color

FFA and AV are oil stability indicators that measure the amount of free fatty acid and alkali and neutralize the free fatty acid in the oil, respectively [30]. A higher value of FFA and AV was observed in unrefined oils, such as CPO, RPO and PKO, compared to refined oils such as RBD and SBO. A lower value in refined oil is expected due to the removal of FFA content during the refining, bleaching and deodorization processes to produce refined oil. FFA are more susceptible to autoxidation compared to esterified fatty acids in triacylglycerol form, making them prooxidants in oils [6,11] and increasing the degree of rancidity in the oils [31]. However, the degree of oxidation is also influenced by other factors, such as fatty acid profiles and the presence of antioxidant compounds.

Palm oils (CPO and RPO) contain high carotenoid content, mainly β-carotene, creating an orange-red oil color. Higher β-carotene content can be observed in CPO and RPO. CPO has an average of 500–700 ppm of carotenoids, mainly β-carotene and α-carotene, which comprise 50% and 30% of the total carotenoids, respectively [32]. However, RBD had an undetectable level of β-carotene content due to CPO’s refining, bleaching and deodorization processes that removed carotenoids from the oil. The PKO extracted from palm kernels contained low levels of carotenoids. Similarly, soybeans contained an undetectable level of β-carotene. The soybean extraction and refining process removed the carotenoids to an undetectable level in the soybean.

A higher L* (lightness) value indicates a darker color. The current study showed higher L* in PKO and CPO than in RPO, RBD and SBO. It could be associated with the physical state of oil at room temperature, in which PKO and CPO are in a semi-solid state, whereas RPO, RBD and SBO are in a liquid state. The semi-solid state is described as a highly viscous oil as the oil starts to solidify at temperatures below the melting point. The a* value was greater in CPO and RPO than in RBD, PKO and SBO. In terms of yellowness, CPO, RPO and PKO showed a higher value of b*. The higher amount of a* in CPO and RPO could be associated with a higher β-carotene content. A higher b* in CPO was contributed by carotenoids present in combination with the oil’s semi-solid state, which enhanced the oil’s yellowness.

### 4.2. Total Phenolics, Flavonoids, Tannin and TBARS of Oils

The TPC was higher in unrefined oils, such as CPO, RPO and PKO, than in refined oils such as RBD and SBO. However, RBD had a higher TPC than SBO, although both were refined oils. The refining process to remove FFA through deacidification and deodorization removes the phenolic compounds from the oil [33]. Higher TBARS in SBO would reflect higher lipid oxidation and a reduction of phenolic compounds. A similar trend can be observed in TFC and TTC, where the less processed oil had a higher value of TFC and TTC. Abdullah et al. [33] found the highest TPC in CPO, followed by RBD, and the least in PKO. The reduction of TPC can be associated with the fractionation processes of CPO and crude PKO to produce RBD and refined PKO. The losses of phenolic compounds are related to absorption by bleaching earth, degradation and volatilization during refining processes [33]. The reduction in the subgroup of phenolic compounds, flavonoids and tannins in refined oils, such as RBD and PKO, can also be related to the losses during the fractionation process.

The TBARS assay measures MDA concentration to indicate the oxidation state of the oil. The current study observed the highest TBARS value in SBO (56.56 µg MDA/g), followed by PKO (31.30 µg MDA/g), RPO (14.81 µg MDA/g) and CPO (10.95 µg MDA/g), with the lowest in RBD (6.568 µg MDA/g). Fatty acid composition could predict the degree of lipid oxidation, in which the presence of USFA, particularly PUFA, is the primary substrate for oil peroxidation [9], which could cause a higher TBARS value in SBO. The SBO in this study had a higher PUFA fraction, at around 41%, compared to low-PUFA palm oils (4 to 5% PUFA) and PKO (1.5% PUFA). The presence of phenolic compounds that act as antioxidants in oil is beneficial in preventing oil oxidation. A higher percentage of TPC, TFC and TTC in CPO, RPO and PKO helped reduce lipid peroxidation, even though the concentrations of FFA and AV were higher than in other oils. Crude oils are reported to be more stable than refined oils due to the suspended and dispersed materials having antioxidant compounds that stabilize the oils and prevent oxidation [34].

### 4.3. Antioxidant Enzymes of Liver

An imbalance of free radical production during metabolic processes and antioxidant defense will result in oxidative stress. The production of antioxidant enzymes by the body cells is vital to balance the production of free radicals with antioxidant defense and prevent damage from the excessive presence of free radicals [35]. In this study, contrary to gene expression, various dietary supplementations of oils affected the concentration of liver antioxidant enzymes. Lower liver SOD and GPX enzyme activity were observed in dietary supplementation with palm oils (CPO, RPO and RBD). Palm oils contribute to the higher deposition of antioxidants, such as carotenoids and vitamin E, in the liver tissues. Karadas et al. [36] reported that the dietary supplementation of vitamin E and carotenoids for 42 days increased the antioxidant accumulation in the liver tissue of broiler chickens. The presence of antioxidants acts on oxidants to prevent oxidation and reduce the production of oxidation products [37]. Thus, a reduction in the formation of oxidants reduces the requirement for antioxidant enzyme production [13,16].

Lower liver SOD and GPX enzyme activity were observed in dietary supplementation with PKO. PKO had a lower concentration of carotenoids and vitamin E than palm oil (CPO, RPO and RBD) and SBO but a higher MCFA. The lower antioxidant enzyme activity of the liver in dietary supplementation of PKO was contributed by MCFA in the PKO. Therefore, it is proposed that MCFA in PKO also acts as an antioxidant and might contribute to lowering oxidative stress. This was proved by Sengupta et al. [15], who reported the contribution of higher antioxidant activity by MCFA in rice bran oil (RBO) enriched with MCFA compared to normal RBO without MCFA. RBO is known to have high USFA (about 80%), contributed by oleic (C18:1) and linoleic (C18:2n-6) acids without the presence of MCFA, and the enriched RBO contains either 14% C8:0 (caprylic), 13.4% C10:0 (capric) or 13.1% C12:0 (lauric acid). In addition, higher inhibition of lipid peroxidation was shown by MCFA-enriched RBO through lower TBARS and conjugated dienes than normal RBO [15]. Sengupta et al. [15] also found that the shorter the carbon chain of MCFA, the higher the antioxidant activity and the greater the prevention of lipid peroxidation. Thus, the MCFA in PKO, similar to the MCFA-enriched RBO, may contribute as antioxidants to reduce oxidants and as the requirement to produce more antioxidant enzymes.

In contrast to palm oils and PKO, a higher liver SOD and GPX enzyme activity was observed in dietary supplementation with SBO. Higher PUFA in the diet of SBO (65% PUFA) may contribute to a higher increment in liver SOD and GPX activity than in palm oils (50% PUFA) and PKO (36% PUFA). A higher PUFA diet may increase the degree of lipid oxidation due to double bonds compared to a lower PUFA diet. Sadžak et al. [38] reported that higher PUFA in the phospholipid bilayer enhances the susceptibility of reactive oxygen species to cause oxidative damage and changes in membrane integrity, such as a reduction in membrane fluidity-induced barrier dysfunction. Many antioxidant enzymes are stress-inducible (GPX, SOD and some selenoprotein-related antioxidant enzymes). The production and activity of enzymes depend on the level of oxidative stress [13,16]. Therefore, the presence of higher antioxidant enzymes in SBO could be associated with the body’s response to protect the body itself against oxidative stress and may not cause detrimental effects on the birds.

### 4.4. Antioxidant Enzymes of Serum

Antioxidant enzyme activity is measured in blood serum to evaluate the body’s antioxidant status. Dietary supplementation of different oils did not affect serum SOD but affected serum GPX and CAT. Generally, serum GPX and CAT were lower with dietary supplementation of palm oils (CPO, RPO and RBD) and PKO compared to SBO. GPX serum was lower in PKO and CAT serum was lower in CPO and RPO compared to SBO. Like the regulation of antioxidant enzymes in the liver, the presence of lower USFA fractions in palm oils (CPO, RPO and RBD) and PKO contributed to a reduction in the degree of oxidation, reducing the need for higher production of antioxidant enzymes. Sadžak et al. [38] reported that higher PUFA in the phospholipid bilayer enhances the susceptibility of reactive oxygen species to cause oxidative damage, and changes in membrane integrity, such as a reduction in membrane fluidity, induce barrier dysfunction. In addition, PKO contains a high amount of MCFA, an excellent antioxidant [15] that may reduce oxidative stress in the birds.

In addition to fatty acid profiles, the presence of carotenes (CPO and RPO) and vitamin E (CPO, RPO and RBD) may serve as antioxidants and contribute to the reduction of oxidative stress. Karadas et al. [36] reported that including vitamin E and carotenoids (lutein, lycopene, canthaxanthin, apo-ester, lutein and β-carotene) in the diet of broiler chickens increased the antioxidant accumulation in the liver tissue. Karadas et al. [36] also discovered that dietary supplementation of vitamin E alone enhanced the total carotenoids in blood plasma. The presence of antioxidants acts on oxidants to prevent oxidation and reduce the production of oxidation products [37]. Thus, a reduction in oxidant formation reduces the requirement for antioxidant enzyme production [13,16].

### 4.5. Antioxidant Enzymes of Intestinal Mucosa

The intestinal tract is crucial in the digestion and absorption of nutrients and is prone to exposure to oxidants in the feed, causing oxidative stress and tissue damage [39]. The current study explored the influence of various oils on the regulation of antioxidant enzymes based on the concentration of enzymes in the jejunal mucosa. Contrary to the expression of genes, the concentration of GPX and CAT at the jejunal mucosa was not affected by different oils, except for SOD. Furthermore, the SOD enzymes showed similar activity between dietary supplementations of oils, as indicated by similar SOD enzyme activity between CPO, RBD, PKO and SBO.

RPO showed the lowest SOD at the jejunal mucosa, despite not being different from CPO and PKO. The SOD enzyme is essential in the antioxidant system because it scavenges the superoxide anion and converts it to oxygen and hydrogen peroxide for the subsequent action of other antioxidant enzymes, such as GPX and CAT, to convert hydrogen peroxide. Therefore, the presence of carotenoids and vitamin E in the CPO and RPO may contribute to the increase in antioxidant capacity in the mucosal tissues, and the requirement for higher production of antioxidant enzymes may be less. Karadas et al. [36] reported that the dietary supplementation of vitamin E and carotenoids increased the antioxidant accumulation in the liver tissue of broiler chickens. In this study, higher β-carotene in feed containing CPO and RPO led to a greater accumulation of β-carotene in the yolk and liver [2].

It is well known that natural antioxidants, such as carotenoids and vitamin E, have antioxidative properties that prevent oxidation [40,41]. The presence of antioxidants acts on oxidants to avoid oxidation and to reduce the production of oxidation products [37]. CPO and RPO contain natural antioxidants in the form of carotenoids and vitamin E (tocopherols and tocotrienols) that contribute to the prevention and reduction of the oxidation process. Reduction in SOD in jejunal mucosa could also be related to the lesser requirement of antioxidant enzyme production to reduce the oxidation activity; as Surai [16] and Surai et al. [13] stated, the production of antioxidant enzymes is in response to the levels of oxidative stress.

### 4.6. Antioxidant Enzyme Gene Expression of Liver and Intestinal Mucosa

The quantification of antioxidant genes was conducted to give better insight into the effects of different oils on antioxidant enzyme production at the molecular level. The current study disclosed the similarity in liver CAT, SOD and GPX gene expression in laying hens fed different oils. However, the effects of different dietary supplementations of oils on gene expression were observed in jejunal mucosa. Jejunal mucosa SOD, GPX and CAT were highly upregulated in SBO compared to palm oils (CPO, RPO and RBD) and PKO. Furthermore, dietary supplementation with oil with a higher PUFA increased the jejunal mucosa expression of the three significant antioxidant enzymes compared to MCFA-rich and palmitic acid (C16:0)-rich oils. The fatty acid composition deposited in the body tissue reflected the fatty acid composition in the diet. Higher PUFA concentrations in the serum, liver and yolk were found to be higher in SBO than in palm oils (CPO, RPO and RBD) and PKO.

However, the presence of PUFA in the phospholipid bilayer increases the susceptibility of reactive oxygen species to cause oxidative damage, and changes in membrane integrity, such as a reduction in membrane fluidity, induce barrier dysfunction [38]. The higher presence of natural antioxidants in palm oils, such as carotenoids and vitamin E (particularly tocotrienols), and a high fraction of MCFA in PKO may reduce oxidative stress and the requirement to produce antioxidant enzymes. Many antioxidant enzymes, such as GPX, SOD and some selenoprotein-related antioxidant enzymes, are stress-inducible. The production and activity of antioxidant enzymes depend on the level of oxidative stress [13,16].

Tan et al. [42] reported that induced oxidation of SBO increased the peroxide value, p-anisidine value, secondary lipid peroxidation products and decreased the vitamin E concentration. Hence, feeding highly oxidized SBO to broiler chickens reduced the total antioxidant capacity in the liver and increased the TBARS in the jejunal mucosa. Contrary to our findings, Tan et al. [42] found that SOD was decreased in the liver and ileal mucosa. Induced oxidation of SBO by heating may cause the oil to be highly oxidized and contribute to reductions in antioxidant capacity and enzymes.

## 5. Conclusions

In summary, free fatty acid and acid values were present in a significant amount in lesser refined oils, such as CPO, RPO and PKO, compared to RBD and refined SBO. The CPO and RPO had higher carotenoids in the form of β-carotene, which was reflected in the redness and yellowness of the oils. High antioxidant compounds, such as TPC, TFC and TTC, are present in palm oils and PKO and reduce the oxidation of oils. Refined oil had lower FFA and AV, and the presence of antioxidant compounds in the oil helped to reduce the degree of oxidation and the presence of oxidation products. The liver, serum and jejunal mucosal antioxidant enzyme activities were lower in palm oils, particularly CPO and RPO. The jejunal mucosa antioxidant enzymes were downregulated in palm oils and PKO, but there was no difference between oils in liver antioxidant enzymes. It is associated with a lower requirement for producing antioxidant enzymes in dietary oils with higher antioxidant compounds and capacity. It is suggested that dietary supplementation with CPO is cost-effective and preferred because of the high content of antioxidant compounds, and it contributes to better antioxidant protection in laying hens at a lower price compared to other oils.

## Figures and Tables

**Figure 1 animals-13-02245-f001:**
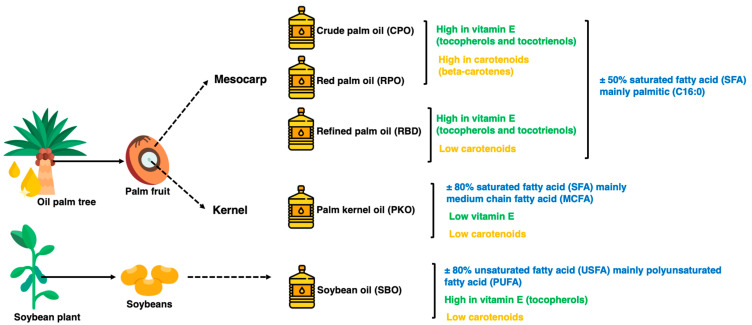
The physical state of different oils at room temperature.

**Figure 2 animals-13-02245-f002:**
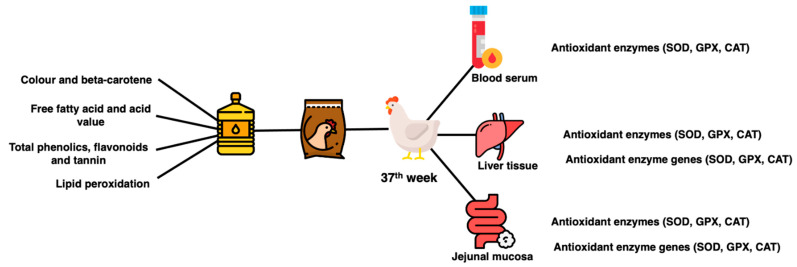
Summary of sample collection and analysis.

**Figure 3 animals-13-02245-f003:**
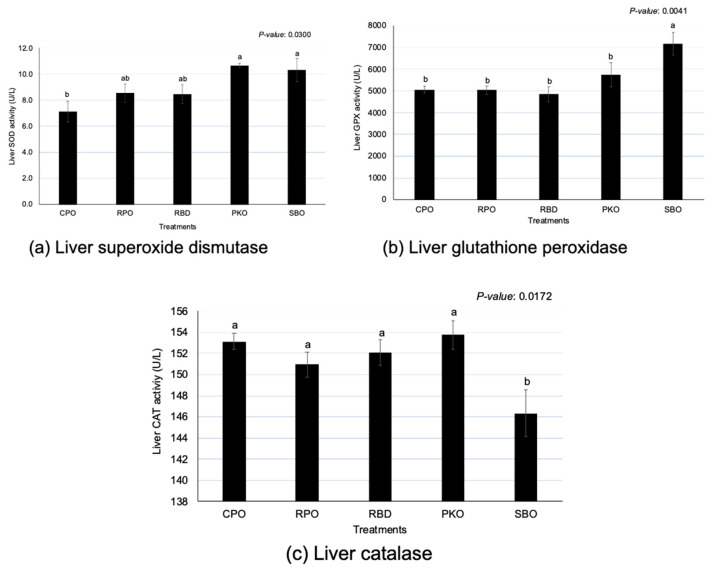
Liver superoxide dismutase, glutathione peroxidase and catalase in laying hens fed different oils. CPO: crude palm oil, RPO: red palm oil, RBD: refined palm oil, PKO: palm kernel oil, SBO: soybean oil, SOD: superoxide dismutase, GPX: glutathione peroxidase, CAT: catalase. ^a,b^ Means with different superscripts on error bars between oils depict significant differences (*p* < 0.05).

**Figure 4 animals-13-02245-f004:**
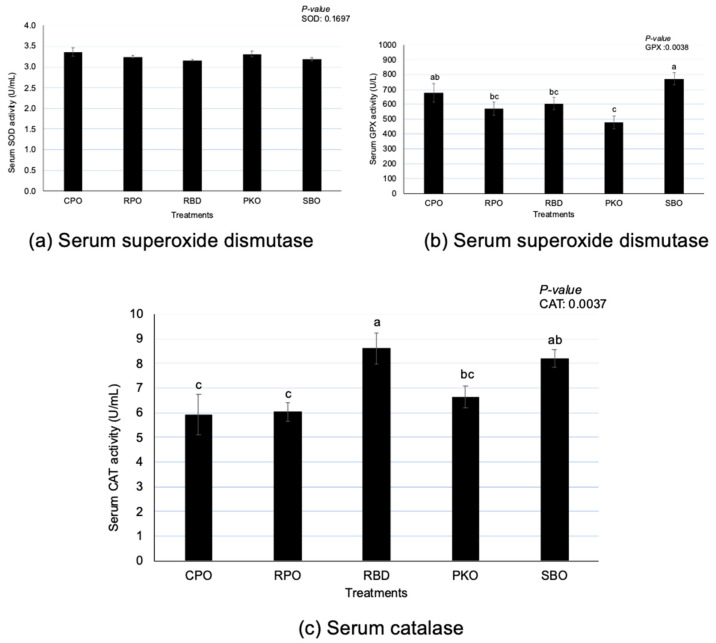
Serum superoxide dismutase, glutathione peroxidase and catalase in laying hens fed different oils. CPO: crude palm oil, RPO: red palm oil, RBD: refined palm oil, PKO: palm kernel oil, SBO: soybean oil, SOD: superoxide dismutase, GPX: glutathione peroxidase, CAT: catalase. ^a,b,c^ Means with different superscripts on error bars between oils depict significant differences (*p* < 0.05).

**Figure 5 animals-13-02245-f005:**
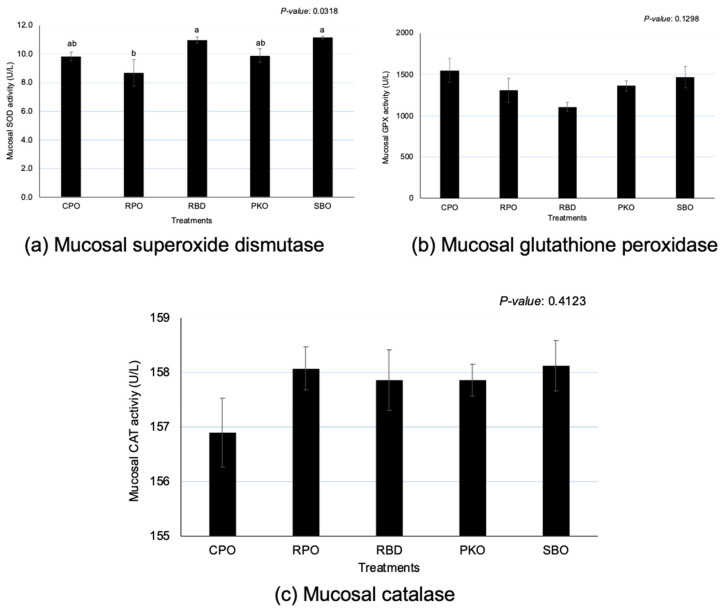
Jejunal mucosa superoxide dismutase, glutathione peroxidase and catalase in laying hens fed different oils. CPO: crude palm oil, RPO: red palm oil, RBD: refined palm oil, PKO: palm kernel oil, SBO: soybean oil, SOD: superoxide dismutase, GPX: glutathione peroxidase, CAT: catalase. ^a,b^ Means with different superscripts on error bars between oils depict significant differences (*p* < 0.05).

**Table 1 animals-13-02245-t001:** The fatty acid profiles of oils.

	CPO	RPO	RBD	PKO	SBO
C8:0	0.019	0.014	0.010	2.680	0.009
C10:0	0.021	0.019	0.015	2.776	0.006
C12:0	0.521	0.452	0.392	44.61	0.422
C14:0	0.900	0.856	0.843	16.20	0.152
C16:0	55.54	51.85	49.80	13.44	22.37
C18:0	3.833	3.812	3.547	2.373	5.529
C18:1	34.37	37.71	40.76	16.71	29.84
C18:2n-6	4.606	5.087	5.564	1.420	37.22
C18:3n-3	0.126	0.139	0.069	0.003	4.302
SFA	60.87	57.04	53.58	81.86	28.54
USFA	39.13	42.96	46.42	18.14	71.46
MUFA	34.39	37.73	40.79	16.72	29.88
PUFA	4.735	5.230	5.637	1.424	41.58

CPO: crude palm oil, RPO: red palm oil, RBD: refined palm oil, PKO: palm kernel oil, SBO: soybean oil, SFA: saturated fatty acid, USFA: unsaturated fatty acid, MUFA: monounsaturated fatty acid, PUFA: polyunsaturated fatty acid. Source: Izuddin et al. [3].

**Table 2 animals-13-02245-t002:** Formulation and nutrient composition of feeds containing different oil sources.

	CPO	RPO	RBD	PKO	SBO
Raw materials (%)					
Corn	48.90	48.90	48.90	48.90	48.90
Soybean meal	28.00	28.00	28.00	28.00	28.00
Wheat pollard	8.000	8.000	8.000	8.000	8.000
Crude palm oil	3.000	-	-	-	-
Red palm oil	-	3.000	-	-	-
Refined palm oil	-	-	3.000	-	-
Palm kernel oil	-	-	-	3.000	-
Soybean oil	-	-	-	-	3.000
DL-Methionine	0.300	0.300	0.300	0.300	0.300
MDCP	2.300	2.300	2.300	2.300	2.300
Calcium carbonate	8.350	8.350	8.350	8.350	8.350
Choline chloride	0.200	0.200	0.200	0.200	0.200
Salt	0.350	0.350	0.350	0.350	0.350
Mineral mix	0.200	0.200	0.200	0.200	0.200
Vitamin mix	0.200	0.200	0.200	0.200	0.200
Antioxidants	0.100	0.100	0.100	0.100	0.100
Toxin binder	0.100	0.100	0.100	0.100	0.100
TOTAL	100.0	100.0	100.0	100.0	100.0
Calculated nutrient (%)					
ME (kcal/kg)	2790	2790	2790	2790	2790
Crude protein	17.17	17.17	17.17	17.17	17.17
Ether extract	4.980	4.98	4.98	4.98	4.98
Crude fibre	3.800	3.80	3.80	3.80	3.80
Calcium	4.000	4.00	4.00	4.00	4.00
Total phosphorus	0.840	0.84	0.84	0.84	0.84
Avail. phosphorus	0.460	0.46	0.46	0.46	0.46
Methionine	0.581	0.581	0.581	0.581	0.581
Lysine	0.933	0.933	0.933	0.933	0.933

CPO: crude palm oil, RPO: red palm oil, RBD: refined palm oil, PKO: palm kernel oil, SBO: soybean oil, ME: metabolizable energy, MDCP: mono dicalcium phosphate. The components of the vitamin and mineral mix were as described by Azizi et al. [27].

**Table 3 animals-13-02245-t003:** The information of target and housekeeping genes.

Target Gene	Primer Sequence	Product Size (bp)	Accession No.
GAPDH F	CTGGCAAAGTCCAAGTGGTG	275	NM_204305.1
GAPDH R	AGCACCACCCTTCAGATGAG		
SOD F	CACTGCATCATTGGCCGTACCA	224	NM_205064.1
SOD R	GCTTGCACACGGAAGAGCAAGT		
GPX F	GCTGTTCGCCTTCCTGAGAG	118	NM_001277853.2
GPX R	GTTCCAGGAGACGTCGTTGC		
CAT F	TGGCGGTAGGAGTCTGGTCT	112	NM_001031215.2
CAT R	GTCCCGTCCGTCAGCCATTT		

GAPDH: glyceraldehyde 3-phosphate dehydrogenase, SOD: superoxide dismutase, GPX: glutathione peroxidase, CAT: catalase, TNFα: tumor necrosis factor-alpha, TLR4: toll-like receptor 4, IL1B: interleukin 1-beta, IL6: interleukin 6, IL10: interleukin 10, IL17: interleukin 17, CLDN1: claudin 1, OCLD: occluding, ZO1: zonula occludens-1, SIGA: secretory immunoglobulin 1, TGFB1: tumor growth factor-beta 1, MUC2: mucin 2.

**Table 4 animals-13-02245-t004:** The free fatty acids, acid value, β-carotene and color profile of oils.

	CPO	RPO	RBD	PKO	SBO	SEM	*p*-Value
FFA (%)	8.107 ^b^	8.787 ^a^	2.05 ^d^	4.610 ^c^	0.770 ^e^	1.864	<0.0001
AV (mg KOH/g)	17.75 ^b^	19.25 ^a^	4.490 ^d^	10.09 ^c^	1.680 ^e^	0.517	<0.0001
β-carotenes (µg/mL)	4.073 ^a^	3.800 ^a^	0.000 ^b^	0.032 ^b^	0.000 ^b^	38.44	<0.0001
L*	31.51 ^b^	16.83 ^d^	18.02 ^c^	41.28 ^a^	18.02 ^c^	1.984	<0.0001
a*	25.61 ^a^	21.10 ^b^	−2.642 ^e^	1.584 ^c^	−1.726 ^d^	2.463	<0.0001
b*	49.42 ^a^	27.02 ^c^	10.24 ^d^	43.63 ^b^	4.746 ^e^	3.635	<0.0001

CPO: crude palm oil, RPO: red palm oil, RBD: refined palm oil, PKO: palm kernel oil, SBO: soybean oil, FFA: free fatty acid, AV: acid value, KOH: potassium hydroxide, L*: lightness (0–100 darker to lighter), a*: (+) redness, (−) greenness, b*: (+) yellowness. ^a,b,c,d,e^ Means with non-identical superscripts across the row indicate significant differences (*p* < 0.05).

**Table 5 animals-13-02245-t005:** Total phenolics, flavonoids, tannin and TBARS of oils.

	CPO	RPO	RBD	PKO	SBO	SEM	*p*-Value
TPC (µg GA/mL)	333.8 ^a^	321.1 ^a^	239.0 ^b^	122.2 ^c^	11.54 ^d^	33.21	<0.001
TFC (µg R/mL)	232.9 ^a^	162.8 ^b^	65.77 ^d^	111.5 ^c^	36.98 ^e^	18.74	<0.001
TTC (µg C/mL)	569.7 ^a^	574.3 ^a^	129.7 ^c^	285.6 ^b^	82.27 ^d^	56.19	<0.001
TBARS (µg MDA/g)	10.95 ^c^	14.81 ^c^	6.568 ^d^	31.30 ^b^	56.56 ^a^	4.912	<0.001

CPO: crude palm oil, RPO: red palm oil, RBD: refined palm oil, PKO: palm kernel oil, SBO: soybean oil, TPC: total phenolic content, TFC: total flavonoid content, TTC: total tannin content, TBARS: thiobarbituric acid reactive substance, MDA: malondialdehyde, GA: gallic acid, R: rutin, C: catechin. ^a,b,c,d,e^ Means with non-identical superscripts across the row indicate significant differences (*p* < 0.05).

**Table 6 animals-13-02245-t006:** Liver and jejunal mucosa antioxidant enzyme gene expression in laying hens fed different oils.

TRT	CPO	RPO	RBD	PKO	SBO	SEM	*p*-Value
Liver							
CAT	2.567	1.845	2.576	1.632	1.000	0.267	0.304
SOD	1.162	0.971	0.931	1.055	1.000	0.066	0.873
GPX	0.427	0.466	0.832	0.744	1.000	0.093	0.238
Mucosa							
SOD	0.112 ^c^	0.281 ^bc^	0.307 ^b^	0.447 ^b^	1.000 ^a^	0.084	<0.001
GPX	0.050 ^c^	0.116 ^bc^	0.323 ^bc^	0.434 ^b^	1.000 ^a^	0.098	0.001
CAT	0.151 ^c^	0.193 ^c^	0.421 ^bc^	0.678 ^ab^	1.000 ^a^	0.101	0.011

CPO: crude palm oil, RPO: red palm oil, RBD: refined palm oil, PKO: palm kernel oil, SBO: soybean oil, SEM: standard error of means, SOD: superoxide dismutase, GPX: glutathione peroxidase, CAT: catalase. ^a,b,c^ Means with non-identical superscripts across the row indicate significant differences (*p* < 0.05).

## Data Availability

Not applicable.

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
