# Peer review of "Antioxidant Enzyme System Modulation by Dietary Palm Oils, Palm Kernel Oil and Soybean Oil in Laying Hens"

_animals, 2023, doi:10.3390/ani13142245_

Round 1

Reviewer 1 Report

The article has relevance and practical significance. However, there are comments and questions. It is necessary to formulate the purpose of the research more specifically. Is there any data on poultry productivity? How did different oils affect productivity? How cost-effective is it to use this or that type of oil. What is the cost of these oils? In the conclusion, it is necessary to clearly make a recommendation on the use of oils. What kind of oil is preferable?

Author Response

The article has relevance and practical significance. However, there are comments and questions.

Q1. It is necessary to formulate the purpose of the research more specifically.

Author’s response: The purpose of the research is revised to be more specific (last paragraph of introduction section).

Q2. Is there any data on poultry productivity? How did different oils affect productivity?

Author's response: Productivity data for poultry supplemented with the oils exists. The productivity and how the oil affects the productivity is included in the manuscript (first paragraph of introduction section)

Q3. How cost-effective is it to use this or that type of oil. What is the cost of these oils?

Author’s response: The cost and cost-effectiveness of the oils to be used in the poultry diet are included in the manuscript (first paragraph of introduction section).

Q4. In the conclusion, it is necessary to clearly make a recommendation on the use of oils. What kind of oil is preferable?

Author’s response: The recommendation of which oil is preferred is included in the conclusion section.

Reviewer 2 Report

This study investigates the antioxidant properties of crude palm oil (CPO), red palm oil (RPO), refined palm oil (RBD), palm kernel oil (PKO), and soybean oil (SBO). The study aims to evaluate the oxidative stability of these oils by analyzing parameters such as fatty acid composition, carotenoid content, vitamin E content, free fatty acid (FFA) content, acid value (AV), color, and antioxidant properties. Additionally, the dietary effects of these oils on antioxidant enzyme systems in laying hens are examined. The results indicate significant differences among the oils in terms of FFA, AV, β-carotene content, and color. 

The article provides a comprehensive examination of the antioxidant properties of palm oil and soybean oil and their potential dietary effects on antioxidant enzyme systems in laying hens. The study design, experimental methods, and data analysis appear to be appropriate. The article contains relevant information and presents the results clearly. 

The manuscript is acceptable for the publication in the journal but it would be better if some sentences were rephrased and edited to convey the meaning more clearly. 

Author Response

The authors would like to thank reviewer 2 for the recommendation and constructive comments on the manuscript.

 Response to Reviewer 2 comments:

The article provides a comprehensive examination of the antioxidant properties of palm oil and soybean oil and their potential dietary effects on antioxidant enzyme systems in laying hens. The study design, experimental methods, and data analysis appear to be appropriate. The article contains relevant information and presents the results clearly. 

The manuscript is acceptable for publication in the journal but it would be better if some sentences were rephrased and edited to convey the meaning more clearly.

Author's response:

Thank you for the recommendation. The manuscript is revised to improve the sentence and convey its meaning.

Reviewer 3 Report

The study explains the Effect of palm-based and soybean oil on Antioxidant Enzyme System in Laying Hens. The manuscript is well written, but the study is explained without showing the control in the study design. qPCR analysis explained that data was analysed by comparing control groups, but control group data is not found within all results. A control group is a collection of factors that remains constant throughout an experiment. Control groups in animal nutrition help researchers determine the efficacy of new nutrient etc., potential side effects and how certain benefits respond to specific treatments. 

Minor editing of English language required

Author Response

The authors would like to thank reviewer 3 for the recommendation and constructive comments on the manuscript.

 Response to Reviewer 3 comments:

The study explains the Effect of palm-based and soybean oil on Antioxidant Enzyme System in Laying Hens.

The manuscript is well written, but the study is explained without showing the control in the study design. qPCR analysis explained that data was analysed by comparing control groups, but control group data is not found within all results. A control group is a collection of factors that remains constant throughout an experiment. Control groups in animal nutrition help researchers determine the efficacy of new nutrient etc., potential side effects and how certain benefits respond to specific treatments. 

Comments on the Quality of English Language: Minor editing of English language required

Author's response:

In this study, we did not set any control group as a comparison because we were interested in comparing each other (between palm oil, palm kernel oil and soybean oil) and relating them based on the unique characteristics of the oils.

In the case of gene expression, since we use Livak’s method, it requires one group to be a control group to measure the expression of a gene relatively. In this experiment, the soybean oil group was chosen to be the control group as it is an oil, which is not produced from palm fruits.

The English language is revised accordingly.

Round 2

Reviewer 3 Report

Suggested to repeat your experiment to add a Control group in your study which is mandatory when you are comparing two different type of oils/feed/ingredients etc

Moderate editing of English language required

Author Response

(The authors gave the same response as above.)
